**METHOD**

# The ENCODE Imputation Challenge: a critical assessment of methods for cross-cell type imputation of epigenomic profiles

Jacob Matthew Schreiber[1][*][†], Carles A. Boix[2][†], Jin wook Lee[1], Hongyang Li[3], Yuanfang Guan[3], Chun-Chieh Chang[4], Jen-Chien Chang[5], Alex Hawkins-Hooker[6], Bernhard Schölkopf[6], Gabriele Schweikert[7], Mateo Rojas Carulla[6], Arif Canakoglu[8], Francesco Guzzo[8], Luca Nanni[9], Marco Masseroli[8], Mark James Carman[8], Pietro Pinoli[8], Chenyang Hong[10], Kevin Y. Yip[11], Jefrey P. Spence[1], Sanjit Singh Batra[12], Yun S. Song[12,13], Shaun Mahony[14], Zheng Zhang[15], Wuwei Tan[16], Yang Shen[16], Yuanfei Sun[16], Minyi Shi[1], Jessika Adrian[1], Richard S. Sandstrom[17], Nina P. Farrell[18], Jessica M. Halow[17], Kristen Lee[17], Lixia Jiang[1], Xinqiong Yang[1], Charles B. Epstein[18], J. Seth Strattan[1], Bradley E. Bernstein[18], Michael P. Snyder[1], Manolis Kellis[2], William S. Noble[19], Anshul Bharat Kundaje[1,20] and ENCODE Imputation Challenge Participants

[†]Jacob Matthew Schreiber and Carles Boix are co-first authors.

*Correspondence:
jmschreiber91@gmail.com

[1] Department of Genetics, Stanford University, Stanford, CA, USA
Full list of author information is available at the end of the article

### Abstract

A promising alternative to comprehensively performing genomics experiments is to, instead, perform a subset of experiments and use computational methods to impute the remainder. However, identifying the best imputation methods and what measures meaningfully evaluate performance are open questions. We address these questions by comprehensively analyzing 23 methods from the ENCODE Imputation Challenge. We find that imputation evaluations are challenging and confounded by distributional shifts from differences in data collection and processing over time, the amount of available data, and redundancy among performance measures. Our analyses suggest simple steps for overcoming these issues and promising directions for more robust research.

## Introduction

Since their development, high-throughput chromatin profiling assays such as histone ChIP-seq, DNase-seq, and ATAC-seq have proven crucial for deciphering gene regulatory elements and characterizing their dynamic activity states across cell types and tissues (together referred to as "cell types" for the rest of this work). Because each assay makes cell type-specific measurements, these assays must be performed for each cell type of interest separately. However, comprehensively profiling a large collection of cell types with assays targeting diverse attributes of chromatin is prohibitive due to practical constraints on material, cost, and personnel. Hence, even the largest repositories of

epigenomic and transcriptomic data are still incomplete in the sense that they are missing tens of thousands of potential experiments [1–6].

To address this challenge, predictive models for imputing missing datasets have been proposed as an inexpensive and straightforward way to obtain complete draft epigenomes [7–11]. These models leverage the complex correlation structure of signal profiles from available experiments to impute signal for experiments that have not yet been performed. Recently, imputation models have been scaled to impute tens of thousands of experiments [12, 13] spanning dozens of assays in hundreds of human cell types. Although progress has clearly been made in developing imputation approaches, the field has thus far only explored a small portion of the space of potential imputation models. Notably, only one of the five methods surveyed above uses nucleotide sequence as input when making imputations.

We organized the ENCODE Imputation Challenge to encourage active development of imputation models. The challenge consisted of two stages and participants were encouraged to share ideas and reorganize into new teams between stages. In the first stage, participants were ranked based on their ability to impute a fixed validation set consisting of experiments randomly selected from within our data matrix. The second stage also measured imputation performance on a held-out set, but with two crucial differences from the first stage: first, the test data was collected during the challenge to ensure a truly prospective evaluation, and second, the test data was collected almost exclusively for poorly characterized cell types (only three of the 12 cell types in the test set have more than two training experiments).

Our initial expectation was that this challenge would primarily serve as an analysis of the components of imputation models and, ultimately, identify those that worked well. However, we found that fairly evaluating the imputations in the second stage was much more challenging than expected, and so the challenge instead served as an impetus to describe, and correct, distributional shifts in large collections of genomics datasets. Specifically, we found that a distributional shift occurs between the more recently collected paired-end data and the older single-end data available on the ENCODE portal. Although we initially expected that this shift was caused by the higher quality of paired-end data, our investigation revealed that it actually arose because of a minor difference in the deduplication step that significantly altered the signal. Without correcting for this difference, we found that a baseline method outperformed all but two of the submissions using the performance measures defined before the challenge began, and those two submissions only performed marginally better than the baseline. After correction, more than half of the participants outperformed the same baseline.

We identified three key challenges in fairly evaluating imputation methods. First, differences over time in experimental procedure or data processing create distributional shifts across experiments which must be corrected for ensure a fair evaluation, and this correction must be more than a simple rescaling of the signal. This concern is particularly important when dealing with data sources, like the ENCODE Portal that contain data collected over long periods of time. Second, while epigenomic imputation is most useful for cell types with few experiments, previous imputation work was evaluated using k-fold or leave-one-out cross-validation applied to an entire compendium. These evaluation settings over-emphasized the performance on well-characterized cell types

and, unfortunately, good performance on well-characterized cell types is not always an indicator of performance on poorly characterized ones. Third, although designing several performance measures is necessary to capture the many aspects of a high-quality experimental readout, designing these measures without accounting for the first two issues can exacerbate redundancy in the measures, limiting their usefulness. As a relevant example, scale-based measures that are appropriate when the predictions and targets are on the same scale will become increasingly redundant as differences in scale increase. We anticipate that giving proper consideration to these three issues in future works will be crucial for developing imputation methods that perform the best in practice.

Accordingly, this work focuses on characterizing the effect that these issues had on evaluating imputation methods, with the goal of providing guidance on how to fairly evaluate such methods in the future. When collecting a test set, one should ensure that processing steps have been uniformly applied to raw data and that the data have been collected using similar procedures. When differences in processing arise that cannot be undone, we propose handling distributional shifts by using a quantile normalization approach that separately normalizes signal in peaks and signal in background. We also propose a set of new performance measures that focus on orthogonal aspects of imputation performance. Finally, we note that performance not generalizing from well characterized cell types to poorly characterized ones is the expected behavior, and so does not have a simple fix like the other issues do. Rather, this disparity can only be evaluated by explicitly including both well-characterized and poorly characterized cell types in the evaluation. At a higher level, one should ensure that at least one setting used to evaluate their approach matches how they expect the method to have the most impact in practice, namely, on poorly characterized cell types.

## Results

### The ENCODE Imputation Challenge

The ENCODE Imputation Challenge was held in two phases (see "The challenge format" section for a complete description). In the first phase, participants were introduced to the problem, given access to the training and validation data, and were subsequently ranked based on the validation set performance of their respective methods. In the second, primary, phase, each participant was ranked based on their method's ability to impute a held-out test set that they did not have access to. Participants submitted 23 models to the second stage of the challenge. Each group was allowed to submit up to three models to encourage inclusion of unorthodox solutions with at least one submission. As a result, the models encompassed a diverse range of strategies (see Table 1). The models differed primarily along three axes. The first axis was the signal preprocessing, with almost every method further preprocessing the data from the given -log10 signal *p*-values. The second axis was the data sources used to construct input features. Most methods followed previously published methods by only using assay measurements as inputs (denoted "functional" in Table 1). However, five of the methods used nucleotide sequence as input, eight methods used the average activity baseline, and three used Avocado's imputations. The third axis was the manner in which the underlying tensor structure of the data was modeled. Some methods explicitly modeled the data as a tensor

**Table 1** Methodologies of imputation methods. The table lists the modeling strategies and input features used by each of the models, as reported by the teams. The models include k-nearest neighbors (KNN), deep tensor factorization (DTF), autoencoders (AE), convolutional neural networks (CNN), hidden Markov models (HMM), and gradient-boosted decision trees (GBT). The authors of Aug2019Impute and CostaLab v2 did not describe their methods

| Name | Model | Norm | Inputs | | | |
|---|---|---|---|---|---|---|
| | | | Sequence | Functional | Average | Avocado |
| Aug2019Impute | | | | | | |
| BrokenNodes/v2 | KNN | arcsinh | | ✓ | | |
| BrokenNodes v3 | KNN | arcsinh | | ✓ | | ✓ |
| CostaLab v2 | | | | | | |
| CUImpute1/CUWA/ICU | ensemble | arcsinh | | ✓ | ✓ | ✓ |
| Guacamole/Lavawizard | DTF | arcsinh | | ✓ | ✓ | |
| HLYG/v1/v2 | GBT | quantile | ✓ | ✓ | ✓ | |
| imp/imp1 | DTF+AE | Cauchy | | ✓ | | |
| KKT-ENCODE | CNN | arcsinh | ✓ | | | |
| LiPingChun | DTF | arcsinh | | ✓ | ✓ | |
| NittanyLions | KNN | | | ✓ | | |
| NittanyLions2 | KNN | quantile | | ✓ | | |
| SongLab | CNN | log1p | | ✓ | | |
| SongLab2 | HMM | | | ✓ | | |
| SongLab3 | CNN | log1p | | ✓ | ✓ | ✓ |
| UIOWA | CNN | quantile | ✓ | ✓ | | |

(e.g., imp and Lavawizard), whereas other methods only implicitly modeled the structure through rule-based approaches or similarity methods (e.g., the Hongyang Li and Yuan-fang Guan (HLYG) and KNN-based approaches).

An initial inspection of the imputations revealed that most methods captured the general shape of the signal well. Examples drawn from H3K27ac in brain microvascular endothelial cells and DNase-seq in DND-41 cells (Fig. 1A/B, Additional file 1: Fig. S1) suggest two sources of error: the misprediction of a small number of peaks relative to the total number of true peaks and the misprediction of the precise signal value within correctly predicted peaks. Focusing on the misprediction of peaks, we noted that some methods made similar mistakes as the average activity baseline, whereas others made similar mistakes as the Avocado baseline (gray highlights in Fig. 1A/B). Unsurprisingly, methods that used Avocado's imputations as input had the highest genome-wide correlation with Avocado's predictions (it is worth noting that CUImpute1 only used Avocado's imputations for some, but not all, assays). In contrast, methods that explicitly used the average activity did not always exhibit higher correlation with it than other methods (Additional file 1: Fig. S2). This finding suggests that, because the average activity can be directly derived from the training set, many types of models are able to implicitly learn it even when not explicitly trained on it.

Next, we comprehensively evaluated the methods using a battery of performance measures that were specified at the beginning of the challenge (see "Performance measures" section, Additional file 2). We found that performance on these measures depended heavily on the imputed assay (Fig. 1C/D). For instance, most models exhibited four orders of magnitude higher MSE on H3K4me3 than on H3K9me3. However, several

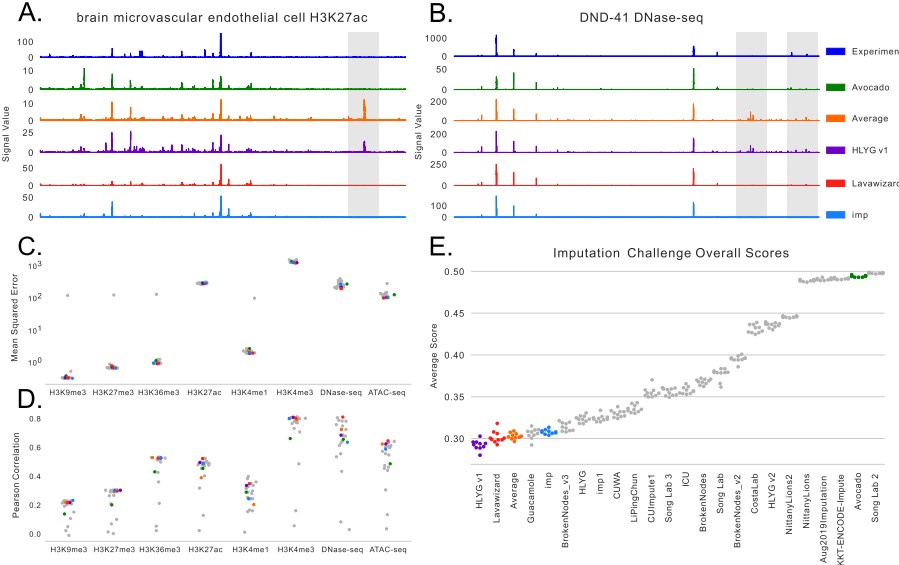

**Fig. 1** Results from the ENCODE Imputation Challenge. **A** The H3K27ac signal for brain microvascular endothelial cells that is observed (in blue), from baseline methods, and from the winning three teams in the challenge. **B** The same as **A** except for DNase-seq signal in DND-41 cells. **C** The average MSE for each method across test set tracks and bootstraps but partitioned by assay type. **D** The same as **C** except for Pearson correlation. **E** The overall score, calculated as described in the "Performance measures" section, across all test set tracks and performance measures shown for each bootstrap for each team. The baseline methods and winners are colored

assays that exhibited the highest MSE also exhibited the highest Pearson correlation, indicating that the scale of MSE across assays is likely more related to the dynamic range of the assay rather than the accuracy of the imputations. Unsurprisingly, a projection of all imputed and experimental tracks clustered predominately by assay type (Silhouette Score = 0.4601), as opposed to by cell type (SS = − 0.4028) or imputation method (SS = − 0.3133, Additional file 1: Fig. S3). Accordingly, we used a rank-based transform to account for differences in dynamic range when calculating global performance measures across experiments (see the "Performance measures" section) to ensure that assays with large dynamic ranges did not dominate the evaluation. After calculating the global performance of each method, we found that there was a gradient of methods that performed increasingly well, and a set of methods that performed relatively poorly (Fig. 1E). The best performing methods, and hence the winners of the challenge, were Hongyang Li and Yuangfang Guan v1 (abbreviated as "HLYGv1") in first place, Lavawizard and Guacamole (two similar methods from the same team) tied for second place, and imp in third place.

Given the diverse modeling strategies of the winning teams, our primary take-away from these results is that there does not appear to be a single key insight that led to good overall performance on the measures used in the challenge. HLYGv1 used nucleotide sequence as input, but so did KKT-ENCODE and UIOWA Michaelson; all three models submitted by Hongyang Li and Yuanfang Guan used gradient boosted trees (GBTs), yet their models exhibited both good and poor performance. However, these results do suggest certain models to be wary of: convolutional neural networks and k-nearest neighbor models underperformed deep tensor factorization (DTF) and GBT models. This is likely

because the similarities used by KNN models are a less sophisticated version of the representations learned by tensor factorization approaches and that the specific structure presented in the data is not well modeled by simple applications of convolutions.

However, when we compared model performance to the baseline methods, we made two important observations. First, almost every team outperformed the Avocado baseline, as one might expect because the participants had access to the Avocado model and predictions during the development process and because the default settings were used for Avocado despite them being tuned for significantly larger amounts of training data. Second, the average activity baseline performed extremely well, coming in third in our ranking and first place in five of the nine performance measures used (Additional file 2). Both of these observations are a reversal from the first round in the challenge, where Avocado outperformed all the participants but almost all the participants outperformed the average activity baseline (Additional file 1: Fig. S4). This reversal in performance between the two baselines is partially because the evaluation setting changed from over-representing well characterized cell types to focusing on poorly characterized ones and, as we will see later, partially due to the performance measures used for the challenge.

### Accounting for distributional shift

A visual inspection of the experimental signal from test set experiments suggested significant distributional differences in peak signal values between the training and test sets for some assays (Fig. 2A). This shift was confirmed by considering the distribution of the training and test set signals within peaks (Fig. 2B). Most obviously, the signal values within H3K4me3 peaks from test set experiments were generally much higher than the signal values within peaks from training and validation set experiments. Although one would expect a locus to exhibit different signal in different cell types because of real biology, one would also expect that the distribution of signal values within peaks across entire experiments would be similar for experiments of the same assay. Because distributional shifts have major ramifications for the scale-based performance measures used in the challenge, we next investigated the source of these distributional differences.

After considering several potential covariates that could explain this distribution shift, including multiple measures of experimental quality (Additional file 1: Fig. S5), we found that the primary driver was a subtle difference in how the test set experiments were processed. By design, the test set experiments were performed during the challenge to ensure a truly prospective evaluation. However, experimental methods have changed in the many years since the training data were collected. Most notably, collecting paired-end data is now the standard approach for ENCODE datasets because the procedure yields higher quality data and is now cheap enough for broad usage; however, almost all of the training set experiments predate this switch and involve single-end data. The processing of single-end and paired-data data is largely similar, but a crucial difference occurs in the deduplication step. Specifically, deduplication of single-end reads using PICARD [14] allows the mapping of only one read start to each position on the genome on each strand. In contrast, deduplication of paired-end data can result in more than one read-start per position on each strand because read-pairs are only removed if the read start of *both* ends are duplicates. Consequently, the number of reads mapping within peaks from paired-end data can be significantly higher than what one would get using

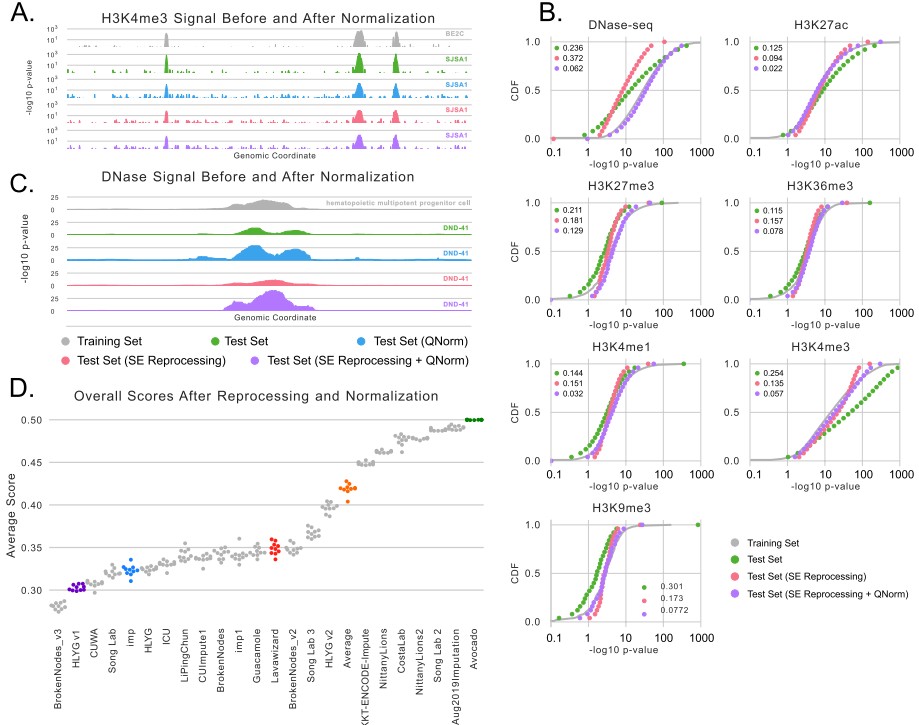

**Fig. 2** Distributional shift and quantile normalization. **A** Experimental signal measuring H3K4me3 in BE2C cells from an unnormalized training set experiment (gray), an unnormalized test set experiment in SJSA1 cells (green), the test set signal after quantile normalization (blue), the test set signal after single-end reprocessing (red), and the test set signal after single-end reprocessing and quantile normalization (purple). **B** Distributions of signal values within peaks in chr16/17 for each reprocessed assay across the unnormalized training set (gray), the unnormalized test set (green), the single-end reprocessed test set (red), and the single-end reprocessed and quantile-normalized test set (purple). The KS statistics between the training set distribution and the test set distributions are shown in the legends and the CDFs are summarized using 25 dots for visualization purposes. **C** An example locus that exhibits a DNase peak in both the training and test sets. **D** A re-scoring of the challenge participants against single-end reprocessed and quantile-normalized test set signal

single-end data. Importantly, the shift is not simply caused by paired-end data being higher quality, as we first explored, but rather differences in the deduplication step.

We confirmed that differences in processing, rather than differences in data quality, explained the distributional shift by reprocessing the paired-end datasets (except for ATAC-seq which requires paired-end data) as single-end data. Specifically, for each paired-end experiment in the test set, we concatenated the FASTQ files of reads from both ends and ran the same single-end processing pipeline that was run on the other single-end experiments in the challenge. We found that the reprocessed data had distributions of peak signal values significantly closer to the training set, as measured by the Kolmogorov-Smirnov (KS) statistic, for four of the histone modification assays including H3K4me3 (Fig. 2B). The remaining two histone modification assays already resembled the training set before reprocessing. However, we found that the distribution of DNase-seq peak signal values had a larger KS-statistic after reprocessing than before. This is likely because 21 of the 38 training set experiments contained paired-end data, which would shift the distribution of signal values in the training set up. Although the most principled next step would be to reprocess all of the experiments used in the challenge

and subsequently re-training and re-evaluating each submission, this analysis was not possible because we only required that the three challenge winners submit code that could retrain their models on new datasets. Given no perfect solution, we chose to continue with the single-end reprocessed test set tracks for our subsequent analyses.

We found that reprocessing the histone modification data significantly reduced the distributional shift but did not perfectly correct it. The remaining differences are likely related to small changes in experimental protocol over time, such as improvements in sequencing technology, antibodies used, and read lengths measured. A general-purpose correction for the remaining differences is to explicitly quantile normalize the data such that the signal values in the testing experiments exhibit the same signal distribution as those in the training experiments. Quantile normalization is powerful because it is a non-linear method, in contrast to min-max or *z*-score scaling, and has been extensively applied to genomics datasets, including those measuring bulk gene expression [15], single-cell RNA-seq [16], and ChIP-seq data when combined with a spike-in reference [17]. We account for differing proportions of the genome exhibiting peaks across cell types by separately quantile normalizing the signal within peaks and the signal in background regions (see the "Quantile normalization" section for details). Finally, because the distribution of signal is significantly different across assays, we apply this quantile normalization to each assay separately. After normalization, we confirmed that the distribution of within-peak test signal values was almost identical to the distribution of within-peak training signal values across all assays (Fig. 2B), even for the DNase-seq experiments.

In theory, one could apply quantile normalization to the original paired-end test set data and, by definition, produce signal values with the same distribution without the need for reprocessing. However, when looking at a representative DNase peak, we found that the reprocessed data was not a simple monotonic transform of the original data (Fig. 2C). Specifically, the paired-end data exhibited a peak shape unlike that observed in the single-end data, and simply quantile normalizing the signal does not fix the differences in shape. More comprehensively, when considering a 10-Mbp region of chr1 on each of the 48 reprocessed experiments, we clearly observed that paired-end data is not a monotonic transformation of single-end data (Additional file 1: Fig. S6). Although the assays associated with activity, such as H3K4me3 and DNase-seq, exhibit Spearman correlations up to 0.938 between the paired-end and single-end processed signals, repressive marks exhibit Spearman correlations as low as 0.037, and the average Spearman correlation across all tracks was only 0.453. Further, even though some assays exhibit high correlation, this value is inflated by the large number of low-signal values and, indeed, the largest variability comes at loci with high signal values.

Moving forward with our method of reprocessing the test data using single-end settings and then quantile normalizing to correct the remaining differences, we next re-scored the originally submitted imputations (Fig. 2D, Additional files 3 and 4). We observed that the number of methods outperforming the average activity baseline increased from two to 16 and that BrokenNodes_v3 rose from sixth place to first place in the rankings. Although HLYGv1 remains within the top three, the other two winners descended in the rankings. This might be explained by HLYGv1 using quantile normalization, albeit a slightly different version than the one we used, during training. Interestingly, many of the methods performed similarly to each other, reinforcing the idea found

in the original challenge that there is not necessarily one way to do imputation. Indeed, the best performing model is a simple KNN-based approach using arcsinh-transformed data and the second best performing model uses gradient-boosting trees on quantile transformed data. Critically, we note that it would not be fair to use these rankings to declare challenge winners because we did not give the teams an opportunity to retrain or tune their methods on the transformed data. Rather, our take-away is that the distributional shift is partially responsible for the good performance of the average activity baseline but does not fully explain it.

### Designing more informative performance measures

Although the measures used in the challenge were devised to rank methods independently for each experiment based on their genome-wide (or across large portions of the genome) performance, this property meant that they ultimately exhibited a high degree of redundancy with each other (Additional file 1: Fig. S7). Essentially, by uniformly weighting all positions along the genome, methods with low genome-wide MSE were likely to have low MSE within promoters, gene bodies, or the top 1% of signal as well. Exacerbating this issue, MSE-based measures were disproportionately confounded by the large distributional shift described in the previous section in comparison to the shape-based measures. Illustrating this, we found that most of the residual—sometimes over 99% in H3K4me3 assays—came at correctly predicted peaks (Additional file 1: Fig S8). Realizing this weakness, we next designed three new types of performance measures that, respectively, reweighted genomic bins based on signal strength, considered multiple experiments simultaneously, and focused on shape within active areas. All evaluations in this section are done against the reprocessed, quantile-normalized test set signal.

#### *Partitioning by signal strength*

A strategy for measuring performance in a complementary way to uniformly weighted genome-wide performance is to explicitly calculate the performance with respect to the magnitude of either the observed or imputed signal (Fig. 3A/B). Rather than being limited by considering only the top 1% bin of signal, such as by using the mse1obs or mse1imp measures, considering all signal bins provides a finer-grained view of model performance. As an example, if the imputations exhibit high accuracy when the imputed signal is high, then one may be confident that predicted peaks are correct when using imputations for which there is no corresponding experimental data; in contrast, if the imputations exhibit low accuracy when the imputed signal is high but higher accuracy when the imputed signal is low, then one might be more skeptical of imputed peak calls but more trusting of regions not called as peaks, e.g., facultative peaks that are not active in the studied cell types. Although any measure can be partitioned by signal magnitude, we focus on accuracy between binarized imputed signal and peak calls for the experimental signal. Accuracy was excluded from the original set of performance measures because the sparsity of peaks can make it difficult to interpret genome-wide; in this setting, we anticipate accuracy to be more valuable in the signal bins where one might reasonably find a peak. Importantly, we did not use rank-based classification measures (e.g.,

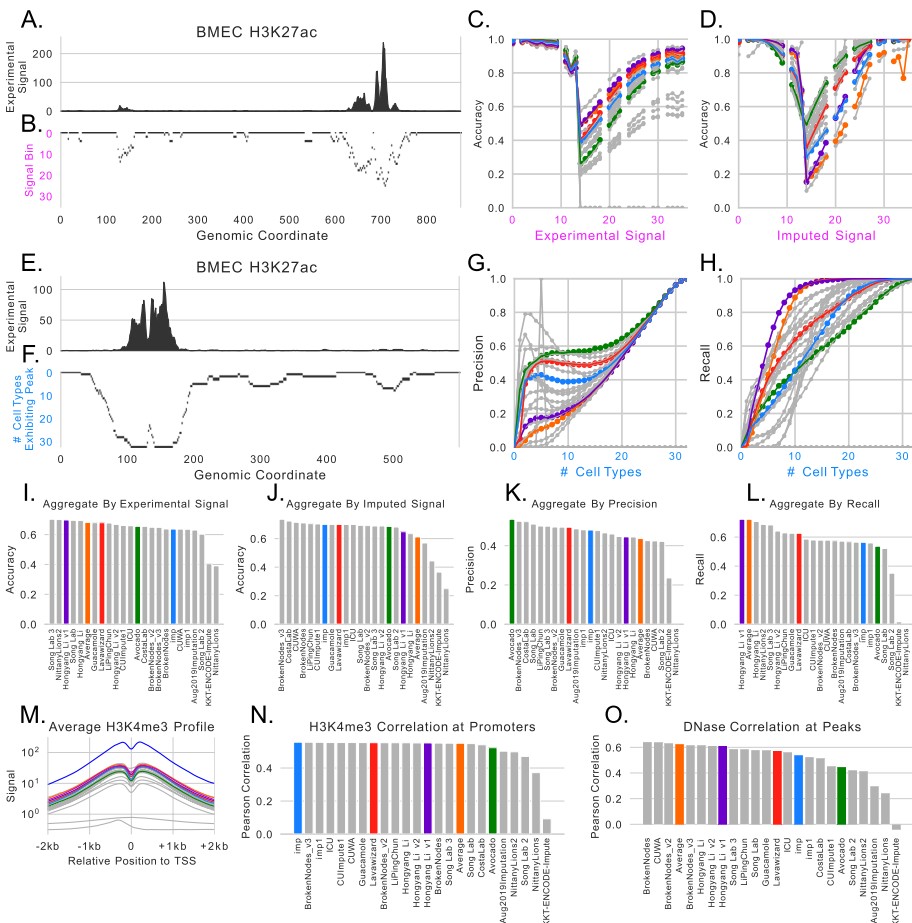

**Fig. 3** Additional performance measures. **A** Experimentally observed signal for H3K27ac in brain microvascular endothelial cells. **B** An example of partitioning the track from A into logarithmically spaced bins (the rows). **C** The accuracy between binarized imputations and MACS2 peak calls for each signal bin when using the experimental signal to define the bins. **D** The same as **C** except using the imputed signal to define the bins. **E** The same as **A** but a different locus. **F** The same as **B** except calculating bins using the number of cell types that each locus exhibits a peak in. **G** The precision of the binarized imputed signal against MACS2 peak calls when evaluated separately for each bin. **H** The same as **G** except the recall instead of the precision. **I** The average area under the curves, calculated as shown in **C**, across all test set tracks for each participant. **J** The average area under the curves calculated as shown in **D** across all test set tracks for each participant. **K** The precision score calculated in the same manner as I/J. **L** The same as **K**, except the recall score. **M** The average H3K4me3 profile of experimental (blue), quantile-normalized (magenta), and imputed signals at strand-corrected promoters. **N** The average Pearson correlation between imputed and quantile-normalized signal across all promoters and H3K4me3 test set tracks. **O** The average Pearson correlation between imputed and quantile-normalized signal across all observed DNase peaks and DNase test set tracks

AUROC or AUPR) here, because once the signal is partitioned by strength, applying a rank-based measure to each bin is less meaningful than when applied genome-wide.

When we partitioned genomic loci based on experimental signal, we found that model performance aggregated across all tracks generally falls into three regimes: (1) when the imputed or experimental signal is low, the accuracy is high, (2) when the imputed or experimental signal is between 1 and 10 the accuracy severely drops, and (3) when the imputed or experimental signal is high, the accuracy returns to being high (Fig. 3C). Although the second regime includes peak calls that may be incorrect due to

ambiguously low signal, it also includes the most difficult to call peaks (thus, the relatively low accuracy) and should be emphasized by performance measures. When focusing on H3K27ac signal in brain microvascular endothelial cells we can also see that rankings flip between the first and second regime (Fig. 3C–D); imp1 and LavaWizard both outperform the average activity and HLYGv1 when the experimental signal is low, but perform significantly lower when the experimental signal is higher.

Interestingly, the ranking of methods is almost reversed when partitioning genomic loci using the imputed signal instead of the experimental signal (Fig. 3D). HLYGv1 and the average activity are among the top performers when partitioning by experimental signal but are among the worst performers when partitioning by imputed signal. An explanation for this flip is that these approaches measure notions of precision and recall, respectively, which have a known trade-off. Because the average activity is essentially a union of peaks across cell types in the training set, it will have a high recall but a low precision. Methods, such as HLYGv1, that rely too heavily on the average activity will exhibit the same tradeoff (Fig. 3C/D).

A straightforward way to condense these curves into a single value for a performance measure is to take the average value across the curve. This value is essentially a re-weighting of genome-wide accuracy that uniformly values each bin of signal values rather than each locus and so will downweight the more common low signal value loci and upweight the less common higher signal values ones. Notably, the winners of the ENCODE Imputation Challenge did not perform the best across all test set experiments when partitioning by either experimental signal or by imputed signal (Fig. 3I/J). Indeed, the top two performers when partitioning by experimental signal (Song Lab 3 and NittanyLions2) came in 12th and 19th respectively in the original evaluation.

### Prediction of facultative peaks

A primary source of error for imputation models comes from loci that exhibit peaks in activity for some, but not all, cell types (i.e., facultative peaks). Evaluating whether the imputations can distinguish between cell types that do and do not exhibit signal at a given locus is crucial for ensuring that the imputations are cell type-specific. However, because traditional genome-wide performance measures treat each experiment independently, they cannot explicitly evaluate this property. To better understand how well these methods can identify what cell types loci are active in, for each assay, we partitioned genomic positions by the number of experiments that exhibit a peak for that assay and then evaluated each partition separately. For example, if a locus exhibited a DNase-seq peak in 3 out of 5 cell types, that locus would be grouped for evaluation with other loci that also exhibited DNase-seq peaks in 3 out of 5 cell types (Fig. 3E/F). This analysis is similar to the one presented by Schreiber et al. [11].

We observe trends that are reminiscent of partitioning loci by signal strength. As the number of cell types that exhibit peaks increases, so too does the precision and recall of the methods (Fig. 3G/H). This indicates that, generally, imputation methods are better at predicting peaks at facultative peaks than they are at predicting cell type-specific activity. Interestingly, we noted that several methods had relatively high precision when the number of cell types the peak was expressed in is low. Given that performance was extremely variable in this regime, we think that focusing on this measure in future studies will be useful

when comparing models. Consistent with the role that the average activity plays as essentially the union of peaks across cell types, we see that it has a low aggregate precision score across all test set tracks but has the second highest aggregate recall score (Fig. 3K/L). Put another way, the average activity is very good at identifying peaks that are common across many cell types but very poor at identifying the cell types that cell type-specific peaks occur in. Somewhat surprisingly, the Avocado baseline had the highest aggregate precision score, but the challenge winners that most resemble it (Lavawizard and imp) did not exhibit the most similar performance.

### *Relative peak shape*

The performance measures that have been proposed so far predominantly involve genome-wide calculations, even if they involve re-weighting loci contributions. An alternate form of performance measure is to focus on specific forms of biochemical activity at loci that are known to be relevant. The MSEProm, MSEEnh, and MSEGene measures attempt to quantify this by focusing on promoters, enhancers, and gene bodies respectively, but measure the performance of all assays at these loci. Next, we investigate two more performance measures that follow the reasoning of Ernst et al. [7] that only specific assays should be measured at these loci.

The first measure evaluates the shape of H3K4me3 signal at promoter regions. This histone modification is known to be enriched at promoter elements and is indicative of active transcription. Further, after correcting for the strand of the promoter, the mark exhibits a distinctive bimodal pattern (Fig. 3M). We reasoned that focusing on the ability to recapture this shape would provide an orthogonal evaluation to the other performance measures proposed so far. We calculated the average Pearson correlation between the imputed signal and the quantile-normalized experimental signal across all gene promoters for all test set tracks measuring H3K4me3. Most of the methods outperformed the average activity baseline but only one of the challenge winners were in the top five according to this measure (Fig. 3N).

The second measure evaluates the shape of DNase signal at observed DNase peaks. We anticipated that recapturing the shape of DNase signal would be more challenging because DNase does not exhibit a pattern that is as consistent as H3K4me3 at promoter regions. Further, the subtle patterns encoded in DNase signal can be useful for deciphering the precise regulatory role that the underlying nucleotide sequence is playing. Consistent with predicting DNase signal being a more challenging task, we found that methods exhibited a wider range of performances than they did with H3K4me3 prediction (Fig. 3O). We also found that only three methods outperformed the average activity baseline. This might initially be counterintuitive, because chromatin accessibility is fairly cell type-specific. However, because this evaluation is limited to observed DNase peaks, methods are not being penalized for incorrectly predicting that non-peak regions are exhibiting peaks. This observation indicates that accessible loci largely retain the shape of their peaks across cell types when binned at 25 bp resolution.

## Discussion

Based on our experience running this challenge, we have several recommendations for the organizers of future challenges involving genomic datasets. First, ensure that participants are compared against naive baselines such as the average activity. Without this

baseline, we might not have identified as easily the distributional shift or the worse performance on sparsely characterized cell types. Second, participants should be required to submit code that can reproduce the training of their models so that more in-depth analysis can be done later. Potentially, the organizers should provide a scaffold that the participants fill in with their own code so that the organizers do not need to decipher each submission to use it properly. Third, organizers should explicitly look for distributional shifts across data splits, and even between pairs of datasets, as a quality control step. For example, paired-end datasets from cancer cell lines can often contain large regional distribution shifts and outliers driven by cell line-specific copy number variation. Even when these shifts are explained by biological processes rather than experimental biases, tailoring an analysis that accounts for these shifts can be an important aspect of a fair evaluation. Finally, organizers should design performance measures that have minimal redundancy with each other, potentially as measured using the average activity before the challenge begins. Naturally, without a singular end-goal in mind it can be difficult to balance the various aspects of performance in a manner that will satisfy everyone, but having redundant performance measures is clearly not helpful.

When the challenge was originally designed, participants were not required to submit working code in order to lower the barrier to entry and allow participants to use their own custom hardware. Although this likely increased participation, it also caused a recurring problem in our later analyses because we could not retrain models on reprocessed data, or on different subsets of data. For example, reprocessing all the data using the single-end settings would likely have been the correct thing to do from a theoretical point of view but was impossible as a practical matter because we did not have the required code. Likewise, we had hypothesized that part of the reason for changes in rankings between the first and second stages (including in our baselines) was because the first stage involved evaluation on a randomly selected held-out test set of experiments, which are biased towards well-characterized cell types, and the second stage explicitly evaluated only poorly characterized cell types. Because we could not re-train the models and evaluate them on cell types giving variable amounts of information, we could not comprehensively pursue this line of inquiry using the challenge data.

An unaddressed, but important, issue is determining the most informative target for imputation methods to predict. The most common target in imputation literature has been the statistical significance from a peak-calling algorithm. Predicting the statistical significance can be more informative than predicting read counts directly because read counts can suffer from unwanted experimental biases and the peak-calling algorithm can explicitly consider a control track. Our challenge setting is consistent with that literature. However, an issue with predicting *p*-values is that fewer tools take those as input than take read counts as input. In fact, performing peak calling using imputations is not obvious because it is unclear that simply thresholding the uncalibrated *p*-values is the correct approach. Potentially, future iterations of the imputation work could involve imputing read counts but allowing models to directly incorporate the control tracks and other covariates such as sequencing depth, single-end or paired-end status, and data quality metrics as well [18]. Although there would be some engineering challenges with such a task, such as designing alternate loss functions or performance measures based on counts, imputation of read counts might be more readily adopted.

As a final consideration, one should explicitly consider the best, context-specific, strategy for normalizing signal from genomic assays. Although one of our major findings was a distributional shift across samples using the same assay driven by endedness, a much more common shift involves assays exhibiting different distributions of signal natively. For example, broad histone marks associated with repression, e.g., H3K9me3, generally exhibit more dispersed reads and hence lower signal *p*-values than narrow histone marks, e.g., H3K4me3. A consequence of these differences is that loss functions and performance measures that rely on scale, such as MSE, will be affected. We noted that all submitted sets of predictions exhibited much higher MSE for H3K4me3 than they did for H3K9me3, but we did not conclude that the models are necessarily worse when predicting H3K4me3 than H3K9me3. Several methods exists for normalizing signal of genomic assays [19–23], but finding the right method for normalizing across both samples and assays will be helpful for fairly training and evaluating any method that operates on massive numbers of tracks.

## Conclusion

A central theme of this work is that evaluating models that rely on large collections of genomic datasets can be more difficult than one might initially expect and, consequently, that results can be confounded even when one does not make any obvious mistakes. In our analysis, we identified three issues that made analysis of imputation models more difficult than we initially thought: distributional differences in the underlying data, previous evaluation focusing on well-characterized cell types and in larger compendia, and performance measures that were either redundant or sensitive to the first two issues. We addressed these issues by proposing a quantile normalization approach that treats peak and background signal separately and proposing new performance measures that were less redundant with each other and covered more aspects of performance than the original measures.

Although the issues we described made the analysis of the results of this challenge more difficult, we made several important findings that we hope will guide the design and analysis of predictive models that rely on genomics data in the future. Specifically, even outside the context of a challenge, being aware of distributional shifts and evaluating a newly proposed model with a wide set of performance measures can help ensure that the model is robust in practice. Further, the difficulties that we faced are not unique to the setting of imputation. Indeed, these issues can affect any model that is trained or evaluated using large collections of publicly available datasets.

## Methods

### The challenge format

We acquired candidate imputation models by hosting the ENCODE Imputation Challenge (https://www.synapse.org/encodeimpute), a public challenge for imputing epigenomic profiles, which began on February 20, 2019, and concluded on August 14, 2019. The challenge evaluated how well predictive models could impute held-out epigenomic experiments using other functional genomic experiments and nucleotide sequence as input (see challenge site for more details). Overall, we acquired 267 datasets from the ENCODE Portal to use as the training set, 45 datasets from the ENCODE Portal to use

as a validation set, and performed 51 new experiments to use as a test set for prospective evaluation (Fig. 4, Additional file 5). These experiments spanned 35 different genomics assays and 51 different cell types. We used chromosomes 1-22 and chrX and excluded chrY and chrM from the challenge.

The challenge was divided into two stages. In the first stage, participants were provided with the training and validation datasets as well as a real-time public leaderboard of performance on the held-out validation set. On this leaderboard, teams BrokenNodes and Hongyang_Li_and_Yuanfang_Guan tied for first place at the conclusion of the first stage (Additional file 1: Fig. S4, Additional file 6). In the second stage, the teams were allowed to re-organize, and participants were encouraged to refine their models using lessons learned from the first stage. Participants were not allowed to use data other than what was provided but were allowed to process it however they chose. The winners of the second stage, and of the entire challenge, were the top three teams based on performance on the held-out prospective test set, which the teams did not have access to.

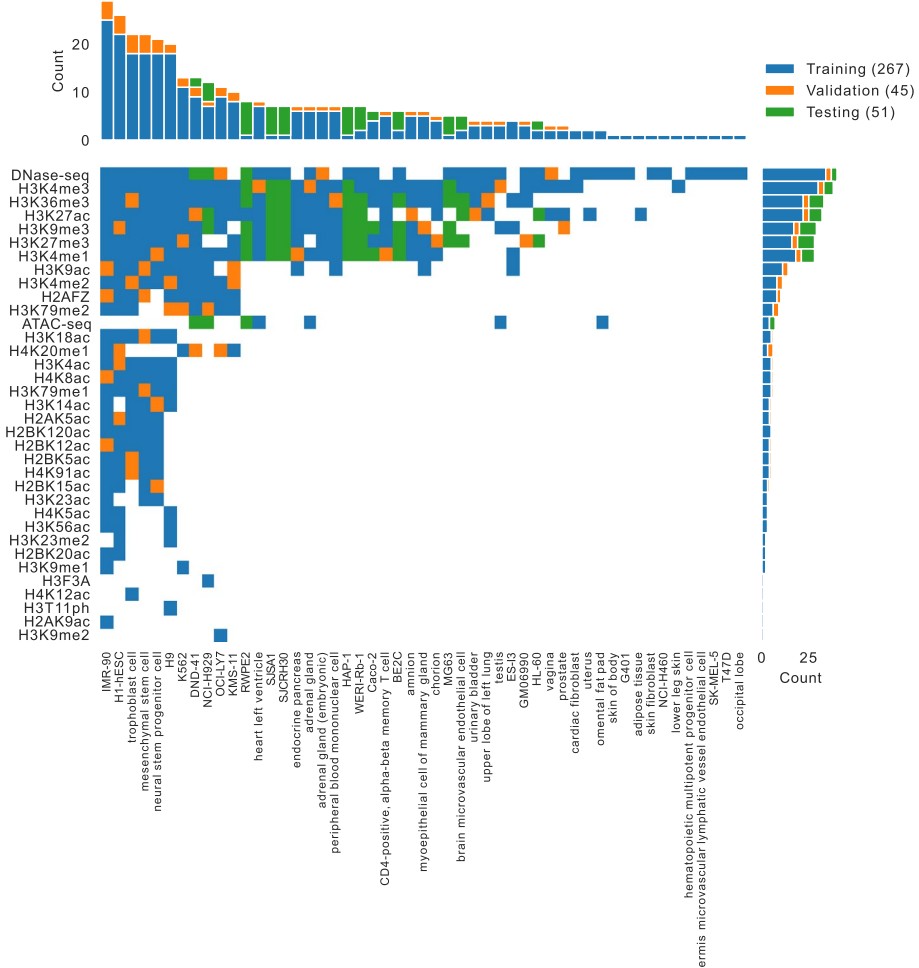

**Fig. 4** The challenge data matrix. The matrix shows the experiments used in the challenge, colored based on whether they were in the training set (blue), the validation set (orange), or the blind test set (green). White squares indicate that an experiment has not yet been performed. The marginal bar plots show the number of experiments in each assay and cell type

The challenge was well attended with 196 people signing up on Synapse. Eight teams submitted results for the first round. After teams merged before the second round, 23 imputation models were submitted. Of these models, only one did not submit the full set of required imputations. Although our method for calculating team rankings as a part of the challenge accounted for missing imputations, our subsequent analyses excluded this model.

**Performance measures**

Prior to the start of the challenge, we specified nine different performance measures to be used in the challenge. Although we provided the signal at basepair resolution, these measures were calculated at 25bp resolution. These performance measures included (1) the genome-wide mean-squared-error (MSE), (2) the genome-wide Pearson correlation, (3) the genome-wide Spearman correlation, (4) the MSE calculated in promoter regions defined as ±2kb from the start of GENCODEv38 annotated genes [24] (MSEProm), (5) the MSE calculated in gene bodies from GENCODEv38 annotated genes (MSEGene), (6) the MSE calculated in enhancer regions as defined by FANTOM5 annotated permissive enhancers [25] (MSEEnh), (7) the MSE weighted at each position by the variance of the experimental signal for that assay across the training set, (8) the MSE at the top 1% of genomic positions ranked by experimental signal (mse1obs), and (9) the MSE at the top 1% of genomic positions ranked by predicted signal (mse1imp). We note that 8 and 9 make a calculation similar to recall and precision, respectively.

We used a multi-stage process, originally developed for the ENCODE Transcription Factor Binding Challenge [26], to aggregate these performance measures into a single score to determine the challenge winners. First, ten equally sized bootstraps were drawn from the pool of all genomic positions, and each of the nine performance measures was calculated for each team on each of the bootstraps for each experiment. For each bootstrap-experiment pair, the scores were converted to rankings across teams for each performance measure, and these rankings were then averaged across performance measures. This resulted in a score for each team in each bootstrap-experiment pair. This score was then converted back into a ranking over teams for each bootstrap-experiment pair. Next, these rankings were aggregated across experiments by calculating $\frac{1}{|E|} \sum_{e \in E} \min(0.5, r_e)$ where $E$ is the set of all experiments, $e$ is an individual experiment, and $r_e$ is a team's ranking on experiment $e$ divided by the number of teams. Finally, a rank was calculated across teams for each bootstrap, and the 90th percentile score, i.e., the second-best bootstrap rank, was used to determine the winners. This procedure is implemented at https://github.com/ENCODE-DCC/imputation_challenge [27].

**Baseline methods**

The methods submitted by the participants were compared to two baseline methods. The first baseline was the average activity, which is a straw-man imputation approach that simply predicts the average training set signal at each position in the genome across all cell types for a given assay type [28]. Consequently, this approach cannot make cell type-specific predictions. However, it represents the simple rule that regions of the genome that always exhibit peaks in signal and that regions of the genome that never exhibit peaks will continue to do so in other cell types. The second baseline was the

Avocado model, using the same model architecture and training procedure described by Schreiber et al [12]. Importantly, this model was not tuned for this dataset—it was applied as-is using the default settings and hyperparameters.

Although we had initially expected that ChromImpute [7] would serve as a baseline in this challenge, for logistical reasons ChromImpute was not applied to the challenge data until well after the challenge concluded. Because the participants did not have access to these predictions, as they did with the other two baselines, we did not include ChromImpute in the original rankings or analysis. However, we have included a ranking of methods that includes ChromImpute in a re-analysis of the challenge participants using six of the measures used to evaluate the original ChromImpute method [7], as a reference (Additional file 1: Table S1/Fig. S9). These measures emphasize the relative distribution of signals, and included Pearson correlation, three measures quantifying percentage overlap between positions exhibiting high signal, and AUC measures for predicting peaks in observed signal from imputed signal values and vice versa. In order to obtain team ranks on these measures, we first ranked each team's prediction for each test track on each measure separately. We then averaged ranks across metrics and re-assigned integer ranks in each track for each team. Each team's final rank was then computed from the average of their predictions' track ranks for the 51 test tracks.

### Quantile normalization

We developed a three-step quantile normalization method for normalizing signal across genomic experiments. Because signal distributions differ significantly across assays, we applied this normalization separately for each assay. Importantly, the normalization is also done separately for signal in peak and background regions (as defined by MACSv2 peak calls for the experiment [29]) to account for peaks spanning differing proportions of the genome across cell types. In the first step, quantiles are derived separately from each training set experiment. That is, if there are $N$ training set experiments, $M_p$ peak quantile bins, and $M_b$ background quantile bins, one would extract the peak quantiles $Q_p \in \mathcal{R}^{N,M_p}$ and the background quantiles $Q_b \in \mathcal{R}^{N,M_b}$. Quantiles are extracted by ranking all signal values for an experiment (in peaks or outside of peaks, respectively), binning those ranks into either $M_p$ or $M_b$ equally sized bins, and assigning to each bin the average signal value from positions within the bin. In the second step, an average is taken across experiments for each quantile bin to construct reference quantiles $R_p \in \mathcal{R}^{M_p}$ and $R_b \in \mathcal{R}^{M_b}$. Finally, $R_p$ and $R_b$ are applied to the test set tracks, with $R_p$ being applied only within signal peaks and $R_b$ being applied only within background regions. Because peak regions are more complex and span a larger range between the minimum and maximum value, i.e., dynamic range, than the background, we set $M_p$ to be 1000 and $M_b$ to be 50. Given that this procedure is designed to combat distributional shift, we note that it should be applied to test set experiments before evaluation.

Although a strength of this approach is that it can handle differing proportions of peaks across cell types, partitioning loci in this manner may introduce minor issues that are worth keeping in mind. First, the same value may map to two different values depending on if it is in a peak or in a background region. Second, if the peak boundaries are extremely conservative, there may be edge artifacts introduced due to the minimum peak quantile being higher than the maximum background quantile.

### Data processing

We processed the DNase and ATAC-seq experiments using a uniform pipeline [30]. First, FASTQ files containing read sequences and quality scores for the training and validation sets experiments were downloaded from the ENCODE Portal, and FASTQs for the test set experiments were acquired from our own experiments. For ATAC-seq experiments (but not DNase-seq), we first trimmed adapters and then mapped reads to the hg38 reference human genome using the Bowtie2 [31] aligner. After mapping, reads were filtered to remove unmapped reads and mates, non-primary alignments, reads failing platform/vendor quality checks, and PCR/optimal duplicates (-F 1804). Reads mapping reliably to more than one location (MAPQ < 30), i.e., multi-mapping reads, were removed. Duplicate reads were then marked with Picard MarkDuplicates [14] and removed. For single-end DNase datasets, a single read was chosen from a set of duplicate reads, whereas for paired-end datasets, read-pairs were chosen if any one of the two reads in the pair was unique. Although this is the standard approach for de-duplicating single-end and paired-end data, this step had unintended consequences for the challenge, which we describe in the "Accounting for distributional shift" section. For ATAC-seq data, 5′ ends of filtered reads on the + and − strand were shifted by + 4 and − 5 bp respectively to account for the Tn5 shift. Reads from biological and technical replicates were merged. We normalized the sequencing depth across datasets by subsampling them to a maximum of 50 million reads (after excluding reads mapping to mitochondria). This number of reads is consistent with best practices for ChIP-seq experiments [32] Although there are several ways to represent the signal from sequencing experiments, e.g., read-counts and fold-change, we chose to use the statistical significance of the fold-change to be consistent with previous imputation literature [7, 10–12]. We used the MACSv2 peak caller to compute the fold-enrichment and statistical significance. MACsv2 was applied to smoothed counts (150 bp smoothing window) of read-starts (5′ ends of reads) at each position in the genome relative to the expected number of reads from a local Poisson-simulated background distribution. We filtered out all peaks that overlapped with the ENCODE Exclusion list consisting of abnormal high signal regions [33]. We provided the genome-wide signal tracks containing the statistical significance of enrichment (i.e., the -log10 *p*-values) at each basepair in the genome. The processing pipeline is open-source and available at https://github.com/ENCODE-DCC/atac-seq-pipeline under the MIT license.

Next, we processed the histone ChIP-seq experiments using the ENCODE processing pipeline [34]. For each experiment, we downloaded FASTQ files from the ENCODE Portal for at least two replicate experiments and a control experiment. All reads were mapped to the hg38 reference human genome using the BWA aligner [35]. After mapping, the process was similar to the ATAC-seq/DNAse-seq pipeline. Reads were filtered to remove unmapped reads and mates, non-primary alignments, reads failing platform/vendor quality checks, and PCR/optical duplicates (-F 1804). Multi-mapping reads (MAPQ < 30) were also removed. Duplicates were identified using Picard MarkDuplicates and subsequently removed, with the same single-end vs. paired-end differences as mentioned for DNase datasets. Reads from the biological and technical replicates were then merged. We normalized the sequencing depth across datasets by subsampling each to a maximum of 50 million reads. We used the MACSv2 peak caller to calculate

fold-enrichment and statistical significance of counts of extended ChIP-seq reads (reads were extended in the $5'$ to $3'$ direction based on the predominant fragment length), relative to the number of extended reads from the control experiment, and filtered out peaks that overlapped with the ENCODE Blacklist [33]. The statistical significance of the enrichment was computed using a local Poisson null distribution whose mean parameter is estimated from the control experiment. For the purposes of this challenge, we provided the genome-wide signal tracks containing the statistical significance of enrichment (i.e., the -log10 *p*-values) at each basepair in the genome. The processing pipeline is open-source and available at https://github.com/ENCODE-DCC/chip-seq-pipeline2 [34] under the MIT license.

### New performance measures

We introduce several new performance measures that can be used to evaluate imputation methods. These measures largely involve partitioning the data into subsets and then calculating standard metrics on the subsets separately. In addition to the continuous-valued experimental signal $X^c \in \mathbb{R}^n$ and the imputations $Y^c \in \mathbb{R}^n$ from a single method being evaluated, we also consider binarized versions of both ($X^b, Y^b \in \{0, 1\}^n$), where $n$ is the length of the genome. The binarized versions of imputations are calculated as $Y^b = Y^c \geq 2$, corresponding to a signal *p*-value of 0.01, and $X^b$ is an indicator for whether each locus is within a MACS2 peak call.

Our first measure involve partitioning the genome according to signal strength. In these cases, we first bin the experimental signal into logarithmic scaled bins of size 0.1 from $10^{-1}$ to $10^{2.5}$. Given an experimental signal bin, we collect the set of loci genome-wide, *loci*, that fall within that bin, and calculate *accuracy*($X^b_{loci}, Y^b_{loci}$) for each method where *accuracy* is a standard implementation of the accuracy measure. We repeated this procedure for each experimental signal bin. Afterwards, we repeated the same procedure using loci derived from imputed signal and binarized experimental signal.

Our next measure involves partitioning the genome according to cell type specificity. The specificity of a locus is defined as the number of cell types that $X^b$ (or $Y^b$) has a value of 1 for a given assay. Specifically, we construct a matrix $A \in \{0, 1\}^{m,n}$ as the stacking of $X^b$ or $Y^b$ vectors across $m$ experiments from the same assay. Then, we can calculate the specificity score as the column sum of this matrix, e.g., $S_j = \sum_{i=0}^{m} A_{i,j}$. Similarly to the previous measure, we can now partition the genome using $S$ by taking all loci that have the same value. We can then calculate *precision*($X^b_{loci}, Y^b_{loci}$) and *recall*($X^b_{loci}, Y^b_{loci}$) using standard implementations of precision and recall.

These performance measures yield one value per bin. To aggregate performance across bins, we simply report the average value for each method. This corresponds, conceptually, to a reweighting of the standard metrics to evenly weight each bin, rather than each locus.

The next performance measure is correlation of H3K4me3 signal at promoter regions, which are defined as $\pm 2kbp$ centered at the starts of genes as defined by the GENCODE v38 annotated gene set. This value is $\frac{1}{|genes|} \sum_{gene \in genes} corr(X^c_{gene}, Y^c_{gene})$ where *gene* is the span of positions defined as a promoter region. Conceptually, it is the average correlation between the experimental and imputed signal across all promoters.

The final performance measure is the correlation of DNase signal at peaks. In contrast with the promoter regions, these peaks are cell type-specific. However, the value is calculated similarly, as $\frac{1}{|peaks|} \sum_{peak \in peaks} corr(X_{peak}^c, Y_{peak}^c)$, with *peaks* being the set of MACS2 peak calls.

## Supplementary Information

---

**Additional file 1.** Supplementary figures and tables for follow-up and supporting analyses.

**Additional file 2.** Performance measures across all ten bootstraps for each of the submitted methods on the challenge test set.

**Additional file 3.** Performance measures across all ten bootstraps for each of the submitted methods on the challenge test set after the test set signal has been quantile normalized.

**Additional file 4.** Performance measures across all ten bootstraps for each of the submitted methods on the challenge test set after the test set signal has been reprocessed to use single-end settings and the signal is subsequently quantile normalized.

**Additional file 5.** All experiments used in the training, validation, and blind test sets for the challenge along with ENCODE accessions.

**Additional file 6.** Performance measures across all ten bootstraps for each of the submitted methods on the challenge round 1 validation set.

**Additional file 7.** Review history.

---

### Acknowledgements

We would like to thank Alan Min for providing feedback on a draft of the manuscript, Oana Ursu, for suggesting analyses, and Jason Ernst for providing manuscript feedback and ChromImpute imputations on the challenge data.

### Peer review information

### Review history

The review history is available as Additional file 7.

### Authors' contributions

A.K., C.B., J.S.S., M.K., and W.S.N. designed the challenge. J.L. processed the data for the challenge. J.L. and J.S.S. provided technical support for the challenge. A.K., W.S.N., J.S., and C.B. ran the challenge. J.S. and C.B. manually validated the challenge winners. J.S. designed and performed the subsequent analyses after the challenge concluded and wrote the manuscript. C.B., A.K., and W.S.N. edited the manuscript.
 H.L. and Y.G. participated in the challenge under the team name "Hongyang Li and Yuanfang Guan." C.C. and J.C. participated in the challege under the team names "LiPingChun," "Guacamole," and "Lavawizard." A.H., B.S, G.S., and M.R.C. participated in the challenge under the team names "imp" and "imp1." A.C., F.G., L.N., M.M., M.J.C., and P.P. participated in the challenge under the team name "BrokenNodes." C.H. and K.Y.Y. participated in the challenge under the team names "CUImpute1," "CUWA," and "ICU." J.P.S., S.S.B., and Y.S.S. participated in the challenge under the team name "Song Lab." S.M. and Z.Z. participated in the challenge under the team name "NittanyLions." W.T., Y.S., Y.S., and Y.S. participated in the challenge under the team name "KKT-ENCODE."
 M.S. and J.A. and R.S. and N.F. and J.H. and K.L. and L.J. and X.Y. and M.C. performed experiments to create the blind test set used to evaluate the methods. The authors read and approved the final manuscript.

### Funding

This work was supported by several grants. J.S. and A.K. were supported by National Institutes of Health awards R01 HG011466 and U01 HG012069. C.B. was supported by National Institutes of Health award U24 HG009446. C.E. was supported by National Institutes of Health award UM1 HG009390. J.S.S. was supported by National Human Genome Research Institute grant 5U24 HG009397. J.P.S. was supported by National Institutes of Health award 5T32 HG000044-23. L.N. and P.P. were supported by European Research Council advanced grant 693174. M.S. was supported by National Institutes of Health award UM1 HG009442. S.M. and Z.Z. were supported by National Science Foundation CAREER award 2045500. W.T. and Y.S. were suported by National Institutes of Health award R35 GM124952. Y.G. was supported by National Institutes of Health award R35 GM133346. Y.S.S. was supported by National Institutes of Health award R35 GM134922. M.S. was supported by National Institutes of Health award UM1 HG009442. W.S.N. was supported by National Institutes of Health award U24 HG009446. M.R.C. was supported by the University of Cambridge and the Max Planck Institute for Intelligent System. The MPI for Intelligent Systems, including B.S. and A.H., acknowledges funding by the Machine Learning Cluster of Excellence, EXC number 2064/1 - Project number 390727645. R.S. and J.H. were supported by National Institutes of Health award UM1 HG009444. J.C. was supported by a research grant from the Japanese Ministry of Education, Culture, Sports, Science and Technology (MEXT) to the RIKEN Center for Integrative Medical Sciences. G.S. was supported by the Academy of Medical Sciences. J.A. was supported by National Institutes of Health award UM1 HG009442. M.K. was supported by National Institutes of Health awards U24 HG009446 and R01 HG008155.

### Availability of data and materials

All data used in the challenge, imputations from the participants and those used as baselines, and submitted models can be found at https://www.synapse.org/#!Synapse:syn17083203/wiki/.
 The ATAC-seq processing pipeline and ChIP-seq processing pipeline [34] are available under a MIT license. The scoring pipeline [27], which was developed for this work, is also available under a MIT license.

## Declarations

### Ethics approval and consent to participate

Not applicable.

### Consent for publication

Not applicable.

### Competing interests

A.K. is a scientific co-founder of Ravel Biotechnology Inc., is on the scientific advisory board of PatchBio Inc., SerImmune Inc., AINovo Inc., TensorBio Inc. and OpenTargets, is a consultant with Illumina Inc., and owns shares in DeepGenomics Inc., Immuni Inc., and Freenome Inc. M.S. is a cofounder and scientific advisor of Personalis, SensOmics, Qbio, January AI, Fodsel, Filtricine, Protos, RTHM, Iollo, Marble Therapeutics, and Mirvie. He is a scientific advisor of Genapsys, Jupiter, Neuvivo, Swaza, and Mitrix. The remaining authors declare no competing interests.

### Author details

[1]Department of Genetics, Stanford University, Stanford, CA, USA. [2]Computer Science and Artificial Intelligence Laboratory, Massachusetts Institute of Technology,  Cambridge, MA, USA. [3]Department of computational medicine and bioinformatics, University of Michigan, Ann Arbor, MI, USA. [4]Department of Research and Development, DeepSeq. AI, San Francisco, CA, USA. [5]RIKEN Center for Integrative Medical Sciences, Yokohama, Japan. [6]Department of Empirical Inference, Max Planck Institute for Intelligent Systems, Stuttgart, Germany. [7]School of Life Sciences, University of Dundee, Dundee, UK. [8]Department of Electronics, Information and Bioengineering, Politecnico di Milano, Milano, Italy. [9]Department of Computational Biology, University of Lausanne, Lausanne, Switzerland. [10]Department of Computer Science and Engineering, The Chinese University of Hong Kong, Sha Tin, Hong Kong. [11]Sanford Burnham Prebys Medical Discovery Institute, San Diego, CA, USA. [12]Department of Electrical Engineering and Computer Sciences, University of California, Berkeley, Berkeley, CA, USA. [13]Department of Statistics, University of California, Berkeley, Berkeley, CA, USA. [14]Department of Biochemistry & Molecular Biology, Center for Eukaryotic Gene Regulation, Pennsylvania State University, University Park, PA, USA. [15]Department of Statistics, Pennsylvania State University, University Park, PA, USA. [16]Department of Electrical and Computer Engineering, Texas A&M University, College Station, TX, USA. [17]Altius Institute, Seattle, WA, USA. [18]Epigenomics Program, The Broad Institute of MIT and Harvard,  Cambridge, MA, USA. [19]Department of Genome Sciences, University of Washington, Seattle, WA, USA. [20]Department of Computer Science, Stanford University, Stanford, CA, USA.

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

## 