## [**Additional file 7. **Review history. · Genome Biology]

Review History

First round of review

Reviewer 1

Were you able to assess all statistics in the manuscript, including the appropriateness of statistical tests used? Yes.

Were you able to directly test the methods? No.

Comments to author:

The paper reports on an open prediction challenge organized by the ENCODE consortium. The goal was to impute missing cell-type specific epigenomic profiles (open chromatin, histone modifications) from a large corpus of existing data.

Overall, the challenge has not lived up to its promises. Using the initially defined and announced challenge rules, the best performing imputation methods barely exceeded a trivial, non-cell-type specific baseline (Fig 2E). Retroactive reprocessing of the test data, application of new data standardization schemes and definition of new performance measures turned out necessary to get meaningful results on the performance of different imputation models. However, the most significant insights gained from this study relate to the analysis and corrections of the initial design errors, not to the comparative benchmarking of different imputation methods.

One important (though not entirely new) insight gained from this study is that data representing biologically equivalent experiments but generated with different platforms and/or processed with different computational pipelines are not comparable. The phenomenon is referred to in this paper as distributional shift. For the challenge, this means that withheld test data generated with platform B had to be numerically back-converted to match source data given to participants, which were generated with an earlier platform A. This problem is far from trivial because the platform-induced changes are drastic, as exemplified by Fig 3C. There one can see that two signal profiles (green and red) for the same genomic region, and generated from the same raw data, once with single-end and once with paired-end processing. Paired-end processing produces a bimodal distribution (green) whereas single-end produces a single dominant peak (red). The authors propose quantile normalization and systematic single-end processing of all data to alleviate the confounding effects of distributional shifts. Other amendments to the challenge rules concern the performance measures. The latter will be important to benchmarkers of new methods or organizers of a new challenge but not to a broader epigenomics readership.

I have also some criticisms regarding the presentation. Overall, I find the paper quite difficult to read, especially the Methods section and the paragraphs of the Results sections dealing with methodological issues. I cannot claim that I completely understood what the authors have done. Important details are missing. Looking at the challenge website

(/www.synapse.org/encodeimpute) was often necessary to get some clues. Here is an example. The paragraph below is from the challenge web site:

"The long axis corresponds to genomic position and, at its finest resolution, has approximately 3.1 billion entries, corresponding to the 3.1 billion base pairs (bp) of DNA in the human genome. In practice, imputation can be carried out at lower resolution. In the ENCODE Imputation Challenge, we will perform imputation at 25 bp resolution, yielding approximately 124 million entries."

At no place in the Method section is it said that the data were binned at a resolution of 25 bp. Only the last sentence of the Results section (line 542) hints that this was done so. Some technical terms were not introduced or sufficiently explained (example "facultative peaks"). At other places, the language itself lacks clarity, for example:

"Interestingly, we noted that several methods had peaks in precision when the number of cell types the peak was expressed in is low."

I'm not sure whether I understand this sentence.

Overall, I feel that imported insights have been gained from this study, but the results and conclusions should be presented in a different way. A new manuscript should focus more on platform-related biases in data, signal standardization methods and performance measures for benchmarking. These issues should be addressed in a broader context, not limited to imputation. Less weight should be given to the challenge itself. A didactic introduction and comprehensive survey of past and present ChIP-seq, DNase-seq and ATAC-seq processing methods should be included in order to make a new paper accessible and interesting to a broader readership.

Reviewer 2

Were you able to assess all statistics in the manuscript, including the appropriateness of statistical tests used? Yes: Statistics are appropriate.

Were you able to directly test the methods? No.

Comments to author:

In the present manuscript, Jacob Schreiber, Carles Boix and colleagues present the results of the "ENCODE Imputation Challenge" with a focus on the possible pitfalls of benchmarking with

regard to the selection and processing of data and fair evaluation measures. In general, I consider this manuscript highly valuable for the community and future benchmarking initiatives in particular. I especially appreciate the recommendations provided in the Discussion section. Still, I have a couple of minor remarks with respect to the current presentation of methods and results as detailed below.

1. I would greatly appreciate a formal definition of the new performance measures described in section 3.3.1 through 3.3.3. While the general idea of each measure is well described in the text, I do not consider the current description sufficient for reproducing results.

2. In the abstract and introduction the authors mention "distributional shifts" in the data sets. After reading the complete manuscript, I understand that this refers to differences in the distributions beyond, e.g., simple shifts and scaling of the observed counts. The authors might consider to characterize the nature of these "distributional shifts" in more detail in the introduction already.

3. In the introduction (line 88), the authors state that "designing these measures without accounting for the first two issues [differences in data processing, well-characterized vs poorly characterized cell types] can introduce redundancy in the measures". However, from my perspective, redundancies in performance measures (also those used in the challenge) may occur independently of the first two issues.

4. The sentence (line 99) was hard for me to comprehend and the authors might consider to rephrase it: "Finally, we note that performance that does not generalize from well characterized cell types to poorly characterized ones does not have a simple fix like the other issues do."

5. The authors should define abbreviations (mse1imp, mse1obs, MSEProm, MSEEnh, MSEGene) used later in the text already in section 2.2 Performance Measures.

6. The authors should give a proper reference for the "ENCODE Transcription Factor Binding Challenge". (Currently, this is just a weblink.)

7. For comparing challenge submissions to ChromImpute (section 2.3), the authors consider a different set of performance measures (those originally used for the ChromImpute method).

a) I wondered, why ChromImpute could not be assessed using the performance measures of this challenge (or those derived later).

b) The authors state that the final rank was computed as the average rank, whereas a rank product (or geometric mean) would seem a more natural choice from my perspective.

8. Figure S2 is not a figure but rather a table.

9. For quantile normalization (2.4), the authors decided to perform normalization for peaks and background separately. In my naive view, this might introduce artifacts with unclear consequences. For instance, this may alter the separation of signal (peaks) and "noise"

(background). It may also lead to situations where two regions in the genome had identical signal values in the original data, but obtain different values after this normalization procedure.

10. In section 2.4 (line 192), the authors use symbols (calligraphic R , Q_p , Q_b) without clear definition.

11. In section 2.4 (line 200), the authors note that normalization should be applied to test set experiments before evaluation. However, in practical applications, one might want to apply a pre-trained model to new (test) data. Hence, a joint normalization of training and test data might be impossible. How would the authors handle such situations?

12. In section 2.5 (line 220), the authors note that reads are subsampled to a maximum number of 50 million reads. As quantile normalization is performed anyway, I did not grasp the necessity of downsampling (with possible loss of information).

13. The abbreviation "HLYG" (line 267, Table 1) should be introduced (maybe, in line 119).

14. In Figure 2E, I wondered why method "imp" (among top 3) is listed after "Guacamole".

15. In the description of section 3.2 (line 328), I would rather refer to Figure 3B than 3A for the distributional shift.

16. In the caption of Figure 4 (line 48), the text should read "(H) The same as G except [...]"

Dear Dr. Pang,

Thank you for managing the review of our article, “The ENCODE Imputation Challenge: A critical assessment of methods for cross-cell type imputation of epigenomic profiles” (GBIO-D-22-01148). We have made several modifications to our manuscript based on the suggestions of the reviewers. Below, the reviewer’s comments are shown in blue, interleaved with our responses in black, and text from the manuscript as images with red text indicating the modifications.

In our response to Reviewer #1, we included extremely positive reviews from our recent submission of a 2-page abstract of this work to the Machine Learning in Computational Biology (MLCB) 2022 conference (attached). We believe that these reviews demonstrate the strength of this work and the value that this submission would bring to the community at large. To be clear, we are not trying to submit these as alternate reviews, but just providing them as some evidence of the value of the work in its current form.

Regarding your question about whether this is a “Method” or a “Research” article: we do not have a strong opinion either way and leave that to your judgement of which one would mesh better with the style of Genome Biology. We originally submitted it as a “Method” paper because the challenge was intended to allow us to evaluate methods.

Reviewer #1: The paper reports on an open prediction challenge organized by the ENCODE consortium. The goal was to impute missing cell-type specific epigenomic profiles (open chromatin, histone modifications) from a large corpus of existing data.

Overall, the challenge has not lived up to its promises. Using the initially defied and announced challenge rules, the best performing imputation methods barely exceeded a trivial, non-cell-type specific baseline (Fig 2E). Retroactive reprocessing of the test data, application of new data standardization schemes and definition of new performance measures turned out necessary to get meaningful results on the performance of different imputation models. However, the most significant insights gained from this study relate to the analysis and corrections of the initial design errors, not to the comparative benchmarking of different imputation methods.

One important (though not entirely new) insight gained from this study is that data representing biologically equivalent experiments but generated with different platforms and/or processed with different computational pipelines are not comparable. The phenomenon is referred to in this paper as distributional shift. For the challenge, this means that withhold test data generated with platform B had to be numerically back-converted to match source data given to participants, which were generated with an earlier platform A.

Although this description is largely correct, we note that the distributional shift is not caused by anything as major as differences in platform or processing pipeline. Rather, it was caused by a minor computational detail in a single step of a commonly-used processing pipeline that

ostensibly handles both paired- and single-end data. We agree that pointing out distributional shifts between platforms or pipelines may not be entirely new, but we believe that pointing out an overlooked source of distributional shift is important, especially when this minor detail causes such major effects.

This problem is far from trivial because the platform-induced changes are drastic, as exemplified by Fig 3C. There one can see that two signal profiles (green and red) for the same genomic region, and generated from the same raw data, once with single-end and once with paired-end processing. Paired-end processing produces a bimodal distribution (green) whereas single-end produces a single dominant peak (red). The authors propose quantile normalization and systematic single-end processing of all data to alleviate the confounding effects of distributional shifts. Other amendments to the challenge rules concern the performance measures. The latter will be important to benchmarkers of new methods or organizers of a new challenge but not to a broader epigenomics readership.

We believe that a broader epigenomics readership may find these measures useful as well, because it is common to compare experimental tracks of data to measure quality or reproducibility. Any measure that evaluates imputation performance can also be used for these purposes.

I have also some criticisms regarding the presentation. Overall, I find the paper quite difficult to read, especially the Methods section and the paragraphs of the Results sections dealing with methodological issues. I cannot claim that I completely understood what the authors have done. Important details are missing. Looking at the challenge website ([/www.synapse.org/encodeimpute](http://www.synapse.org/encodeimpute)) was often necessary to get some clues. Here is an example. The paragraph below is from the challenge web site:

"The long axis corresponds to genomic position and, at its finest resolution, has approximately 3.1 billion entries, corresponding to the 3.1 billion base pairs (bp) of DNA in the human genome. In practice, imputation can be carried out at lower resolution. In the ENCODE Imputation Challenge, we will perform imputation at 25 bp resolution, yielding approximately 124 million entries."

At no place in the Method section is it said that the data were binned at a resolution of 25 bp. Only the last sentence of the Results section (line 542) hints that this was done so.

We apologize that some details were only present on the challenge website. We have added that detail to the paper.

141 Prior to the start of the challenge, we specified nine different performance measures to be
142 used in the challenge. **Although we provided the signal at basepair resolution, these measures**
143 **were calculated at 25bp resolution.** These performance measures included (1) the genome-
144 wide mean-squared-error (MSE), (2) the genome-wide Pearson correlation, (3) the genome-

We have also added a few other details that were mentioned on the website but were not explicitly mentioned in the text.

119 challenge site for more details). Overall, we acquired 267 data sets from the ENCODE
120 Portal to use as the training set, 45 data sets from the ENCODE Portal to use as a valida-
121 tion set, and performed 51 new experiments to use as a test set for prospective evaluation
122 (Figure 1, Additional File 1). **These experiments spanned 35 different genomics assays and**
123 **51 different cell types. We used chromosomes 1-22 and chrX and excluded chrY and chrM**
124 **from the challenge.**

Some technical terms were not introduced or sufficiently explained (example "facultative peaks").

We have provided an explanation in the text for "facultative peaks."

542 3.3.2 Prediction of facultative peaks

543 A primary source of error for imputation models comes from loci that exhibit **functional**
544 **peaks in** activity ~~in~~ for some, but not all, cell types (i.e., **facultative peaks**). Evaluating

We have also defined "dynamic range."

210 R_b being applied only within background regions. Because peak regions are more complex
211 and span a larger ~~dynamic range~~ **range between the minimum and maximum value, i.e.,**
212 **dynamic range**, than the background, we set M_p to be 1000 and M_b to be 50. Given that

We have clarified what we meant by "ambiguous peak calls."

516 experimental signal is high, the accuracy returns to being high (Figure 4C). Although the
517 second regime includes ~~ambiguous peak calls~~ **peak calls that may be incorrect due to am-**
518 **biguously low signal**, it also includes the most difficult to call peaks (thus, the relatively low
519 accuracy) and should be emphasized by performance measures. When focusing on H3K27ac

We would also be happy to clarify any other terms that are confusing.

At other places, the language itself lacks clarity, for example:

"Interestingly, we noted that several methods had peaks in precision when the number of cell types the peak was expressed in is low."

I'm not sure whether I understand this sentence.

Thank you for pointing out this confusing sentence. We have amended it as follows:

557 methods (Figure 4G/H). This indicates that, generally, imputation methods are better at
558 predicting peaks at facultative peaks than they are at predicting cell type-specific activity.
559 Interestingly, we noted that several methods had ~~peaks in~~ relatively high precision when
560 the number of cell types the peak was expressed in is low. Given that performance was
561 extremely variable in this regime, we think that focusing on this measure in future studies
562 will be useful when comparing models. Consistent with the role that the average activity

Overall, I feel that imported insights have been gained from this study, but the results and conclusions should be presented in a different way. A new manuscript should focus more on platform-related biases in data, signal standardization methods and performance measures for benchmarking. These issues should be addressed in a broader context, not limited to imputation. Less weight should be given to the challenge itself. A didactic introduction and comprehensive survey of past and present ChIP-seq, DNase-seq and ATAC-seq processing methods should be included in order to make a new paper accessible and interesting to a broader readership.

We agree that such a review piece would be valuable to the community. However, such a manuscript would significantly expand the scope of the current work, and we believe that the current submission already makes a valuable contribution to the field. Specifically, we describe and dissect a well-attended challenge in an area of interest to many computational researchers. Our findings are useful not only as a post-mortem for the challenge, but the subsequent analyses serve as a warning for computational researchers studying any sequencing-based experimental data, and our experience serves as a guide for future challenges. We have already been asked to help design a different computational challenge to ensure that it does not make the same mistakes.

As further evidence of the interest in this topic by the general community, we recently submitted an abstract describing this work to the Machine Learning in Computational Biology (MLCB) 2022 conference and received extremely positive reviews, mentioning that the lessons learned from this challenge would be invaluable to share in their current form. We have attached these reviews at the end of this response. As a note, these reviews are only meant to demonstrate that the community is interested in the lessons from this challenge, not to supersede your opinions.

Reviewer #2: In the present manuscript, Jacob Schreiber, Carles Boix and colleagues present the results of the "ENCODE Imputation Challenge" with a focus on the possible pitfalls of benchmarking with regard to the selection and processing of data and fair evaluation measures. In general, I consider this manuscript highly valuable for the community and future benchmarking initiatives in particular. I especially appreciate the recommendations provided in the Discussion section.

We thank the reviewer for their positive assessment of our work and their close reading.

Still, I have a couple of minor remarks with respect to the current presentation of methods and results as detailed below.

1. I would greatly appreciate a formal definition of the new performance measures described in section 3.3.1 through 3.3.3. While the general idea of each measure is well described in the text, I do not consider the current description sufficient for reproducing results.

We have added a new methods section, “2.6 Performance Measures,” describing the methods in more detail.

272 2.6 New Performance Measures

273 We introduce several new performance measures that can be used to evaluate imputation
274 methods. These measures largely involve partitioning the data into subsets and then cal-
275 culating standard metrics on the subsets separately. In addition to the continuous-valued
276 experimental signal $X^c \in \mathbb{R}^n$ and the imputations $Y^c \in \mathbb{R}^n$ from a single method being
277 evaluated, we also consider binarized versions of both $(X^b, Y^b \in \{0, 1\}^n)$, where n is the
278 length of the genome. The binarized versions of imputations are calculated as $Y^b = Y^c \geq 2$,
279 corresponding to a signal p-value of 0.01, and X^b is an indicator for whether each locus is
280 within a MACS2 peak call.

281 Our first measure involve partitioning the genome according to signal strength. In these
282 cases, we first bin the experimental signal into logarithmic scaled bins of size 0.1 from 10^{-1}
283 to $10^{2.5}$. Given an experimental signal bin we collect the set of loci genomewide, *loci*, that
284 fall within that bin, and calculate $accuracy(X_{loci}^b, Y_{loci}^b)$ for each method where *accuracy*
285 is a standard implementation of the accuracy measure. We repeated this procedure for each
286 experimental signal bin. Afterwards, we repeated the same procedure using loci derived
287 from imputed signal and binarized experimental signal.

288 Our next measure involves partitioning the genome according to cell type specificity.
289 The specificity of a locus is defined as the number of cell types that X^b (or Y^b) has a value
290 of 1 for a given assay. Specifically, we construct a matrix $A \in \{0, 1\}^{m,n}$ as the stacking
291 of X^b or Y^b vectors across m experiments from the same assay. Then, we can calculate
292 the specificity score as the column sum of this matrix, e.g. $S_j = \sum_{i=0}^m A_{i,j}$. Similarly to the
293 previous measure, we can now partition the genome using S by taking all loci that have
294 the same value. We can then calculate $precision(X_{loci}^b, Y_{loci}^b)$ and $recall(X_{loci}^b, Y_{loci}^b)$ using
295 standard implementations of precision and recall.

296 These performance measures yield one value per bin. To aggregate performance across
297 bins, we simply report the average value for each method. This corresponds, conceptually,
298 to a reweighting of the standard metrics to evenly weight each bin, rather than each locus.

299 The next performance measure is correlation of H3K4me3 signal at promoter regions,
300 which are defined as $\pm 2kbp$ centered at the starts of genes as defined by the GENCODE
301 v38 annotated gene set. This value is $\frac{1}{|genes|} \sum_{gene \in genes} corr(X_{gene}^c, Y_{gene}^c)$ where *gene* is the
302 span of positions defined as a promoter region. Conceptually, it is the average correlation
303 between the experimental and imputed signal across all promoters.

304 The final performance measure is the correlation of DNase signal at peaks. In contrast
305 with the promoter regions, these peaks are cell type-specific. However, the value is calculated
306 similarly, as $\frac{1}{|peaks|} \sum_{peak \in peaks} corr(X_{peak}^c, Y_{peak}^c)$, with *peaks* being the set of MACS2 peak
307 calls.

2. In the abstract and introduction the authors mention "distributional shifts" in the data sets. After reading the complete manuscript, I understand that this refers to differences in the distributions beyond, e.g., simple shifts and scaling of the observed counts. The authors might consider to characterize the nature of these "distributional shifts" in more detail in the introduction already.

We have added more details to the introduction.

69 and correct, distributional shifts in large collections of genomics data sets. Specifically, we
70 found that a distributional shift occurs between the more recently collected paired-end data
71 and the older single-end data available on the ENCODE portal.~~due to small processing~~
72 ~~differences that have a big effect.~~ Although we initially expected that this shift was caused
73 by the higher quality of paired-end data, our investigation revealed that it actually arose
74 because of a minor difference in the deduplication step that significantly altered the signal.
75 Without correcting for this difference, we found that a baseline method outperformed all
76 but two of the submissions using the performance measures defined before the challenge
77 began, and those two submissions only performed marginally better than the baseline. After

3. In the introduction (line 88), the authors state that "designing these measures without accounting for the first two issues [differences in data processing, well-characterized vs poorly characterized cell types] can introduce redundancy in the measures". However, from my perspective, redundancies in performance measures (also those used in the challenge) may occur independently of the first two issues.

Thank you for pointing this out. We have reworded it to clarify that these differences can exacerbate redundancies in scale-based measures.

89 ones. Third, although designing several performance measures is necessary to capture the
90 many aspects of a high-quality experimental readout, designing these measures without
91 accounting for the first two issues can ~~introduce~~ exacerbate redundancy in the measures,
92 limiting their usefulness. ~~As a relevant example, scale-based measures that are appropriate~~
93 ~~when the predictions and targets are on the same scale will become increasingly redundant~~
94 ~~as differences in scale increase.~~ We anticipate that giving proper consideration to these three
95 issues in future works will be crucial for developing imputation methods that perform the
96 best in practice.

4. The sentence (line 99) was hard for me to comprehend and the authors might consider to rephrase it: "Finally, we note that performance that does not generalize from well characterized cell types to poorly characterized ones does not have a simple fix like the other issues do."

We have amended the text to clarify what we mean.

103 normalizes signal in peaks and signal in background. We also propose a set of new perfor-
104 mance measures that focus on orthogonal aspects of imputation performance. Finally, we
105 note that performance ~~that does not generalize~~ not generalizing from well characterized cell
106 types to poorly characterized ones ~~is the expected behavior,~~ and so does not have a simple
107 fix like the other issues do. Rather, this disparity can only be evaluated by explicitly includ-

5. The authors should define abbreviations (mse1imp, mse1obs, MSEProm, MSEEnh, MSEGene) used later in the text already in section 2.2 Performance Measures.

These performance measures were defined in that section without their abbreviation being provided. This has been fixed.

144 wide mean-squared-error (MSE), (2) the genome-wide Pearson correlation, (3) the genome-
145 wide Spearman correlation, (4) the MSE calculated in promoter regions defined as ± 2 kb
146 from the start of GENCODEv38 annotated genes [14] (**MSEProm**), (5) the MSE calculated
147 in gene bodies from GENCODEv38 annotated genes (**MSEGene**), (6) the MSE calculated in
148 enhancer regions as defined by FANTOM5 annotated permissive enhancers [15] (**MSEEnh**),
149 (7) the MSE weighted at each position by the variance of the experimental signal for that
150 assay across the training set, (8) the MSE at the top 1% of genomic positions ranked by
151 experimental signal (**mse1obs**), and (9) the MSE at the top 1% of genomic positions ranked
152 by predicted signal (**mse1imp**). We note that 8 and 9 make a calculation similar to recall
153 and precision, respectively.

6. The authors should give a proper reference for the "ENCODE Transcription Factor Binding Challenge". (Currently, this is just a weblink.)

A paper for this challenge has not yet been published or submitted to a preprint server. However, the Synapse site hosting the challenge resources has a permanent DOI, all relevant documentation, and is citable.

7. For comparing challenge submissions to ChromImpute (section 2.3), the authors consider a different set of performance measures (those originally used for the ChromImpute method).

a) I wondered, why ChromImpute could not be assessed using the performance measures of this challenge (or those derived later).

b) The authors state that the final rank was computed as the average rank, whereas a rank product (or geometric mean) would seem a more natural choice from my perspective.

Ultimately, this ended up being a somewhat thorny logistical issue that arose in part because this work has been ongoing for around five years now. Briefly, the authors of ChromImpute asked for their method to be included in the paper but only provided the model and these rankings long after the challenge had concluded. Given this, we had to balance the value of an additional baseline from an early method with the fact that the finalized challenge rankings that we had already announced did not include ChromImpute and that, ultimately, the evaluation was confounded by the distributional shift. We decided that it was not worth it to change all of the rankings in the paper just to include a new method when that ranking was not even a true reflection of the relative performance of the methods.

8. Figure S2 is not a figure but rather a table.

We have fixed this error.

9. For quantile normalization (2.4), the authors decided to perform normalization for peaks and background separately. In my naive view, this might introduce artifacts with unclear consequences. For instance, this may alter the separation of signal (peaks) and "noise" (background). It may also lead to situations where two regions in the genome had identical signal values in the original data, but obtain different values after this normalization procedure.

We did not find any evidence of these artifacts playing a major role in our evaluation. Figure 3C provides an example of the quantile normalized signal and does not show any artifactual edges to the normalized peak. We have added some text to warn about these potential issues.

215 Although a strength of this approach is that it can handle differing proportions of peaks
216 across cell types, partitioning loci in this manner may introduce minor issues that are worth
217 keeping in mind. First, the same value may map to two different values depending on if

218 it is in a peak or in a background region. Second, if the peak boundaries are extremely
219 conservative, there may be edge artifacts introduced due to the minimum peak quantile
220 being higher than the maximum background quantile.

Further, we strongly believe that splitting loci into background and peaks is necessary despite these potential concerns. Quantile normalizing an entire experiment without splitting the loci would have massive consequences if the proportion of peaks across cell lines differed, which is commonly observed because of both biology and experimental quality. Essentially, if the quantiles are fit to a track that is 10% peaks and used to normalize a track that is 15% peaks, 33% of the peaks in the track being normalized will be eliminated by definition.

10. In section 2.4 (line 192), the authors use symbols (calligraphic R , Q_p , Q_b) without clear definition.

The calligraphic R is the standard nomenclature for real-valued numbers. We have added a definition for Q_p and Q_b .

201 across cell types. In the first step, quantiles are derived separately from each training set
202 experiment. That is, if there are N training set experiments, M_p peak quantile bins, and
203 M_b background quantile bins, one would extract the peak quantiles $Q_p \in \mathcal{R}^{N, M_p}$ and the
204 background quantiles $Q_b \in \mathcal{R}^{N, M_b}$. Quantiles are extracted by ranking all signal values
205 for an experiment (in peaks or outside of peaks, respectively), binning those ranks into
206 either M_p or M_b equally sized bins, and assigning to each bin the average signal value from
207 positions within the bin. In the second step, an average is taken across experiments for each
208 quantile bin to construct reference quantiles $R_p \in \mathcal{R}^{M_p}$ and $R_b \in \mathcal{R}^{M_b}$. Finally, R_p and
209 R_b are applied to the test set tracks, with R_p being applied only within signal peaks and

11. In section 2.4 (line 200), the authors note that normalization should be applied to test set experiments before evaluation. However, in practical applications, one might want to apply a pre-trained model to new (test) data. Hence, a joint normalization of training and test data might be impossible. How would the authors handle such situations?

There is no “joint normalization.” As we mention in the text, the quantiles are derived from the training data and applied to the test data. If new test data were to become available, the same quantiles can be applied. The text describing this is included in the above response.

12. In section 2.5 (line 220), the authors note that reads are subsampled to a maximum number of 50 million reads. As quantile normalization is performed anyway, I did not grasp the necessity of downsampling (with possible loss of information).

The reviewer is correct that quantile normalization can, in principle, account for the same read depth issue that subsampling reads can. However, because our quantile normalization method was only developed and applied in response to the results of the challenge, we did not know at the start of the challenge that we would be using it. Instead, when originally designing the challenge, we chose to use a simpler and more accepted method of accounting for read depth across genomics experiments.

13. The abbreviation "HLYG" (line 267, Table 1) should be introduced (maybe, in line 119).

We have added the full name of the method alongside the first instance of the abbreviation.

320 tensor structure of the data was modeled. Some methods explicitly modeled the data as
321 a tensor (e.g., imp and Lavawizard), whereas other methods only implicitly modeled the
322 structure through rule-based approaches or similarity methods (e.g., the **HLYG Hongyang**
323 **Li and Yuanfang Guan (HLYG)** and KNN-based approaches).

14. In Figure 2E, I wondered why method "imp" (among top 3) is listed after "Guacamole".

As we mentioned in Section 3.1, "Guacamole" and "Lavawizard" are extremely similar methods from the same team. Because the methods were almost identical, they tied for second place according to our ranking procedure described in Section 2.2. We awarded third place to the next method in the ranking, in part because it seemed unfair to allow one team to submit multiple copies of the same method and win multiple spots.

15. In the description of section 3.2 (line 328), I would rather refer to Figure 3B than 3A for the distributional shift.

We have altered the text to specify what, specifically, we were visually inspecting. We think that Figure 3A is the correct figure because it was looking at the plotted tracks themselves that led us to believe there was a distributional shift. Although we confirmed this by looking at the CDFs (the next sentence), we became suspicious by looking at the tracks themselves.

381 3.2 Accounting for distributional shift

382 A visual inspection of ~~the test set experiments revealed~~ the experimental signal from test
383 set experiments suggested significant distributional differences in peak signal values be-
384 tween the training and test sets for some assays (Figure 3A). ~~This shift was confirmed by~~
385 ~~considering the distribution of the training and test set signals within peaks (Figure 3B).~~
386 Most obviously, the signal values within H3K4me3 peaks from test set experiments were
387 generally much higher than the signal values within peaks from training and validation set
388 experiments(~~Figure 3B~~) . Although one would expect a locus to exhibit different signal in

16. In the caption of Figure 4 (line 48), the text should read "(H) The same as G except [...]"

We have fixed this typo.

Second round of review

Reviewer 1

The authors have taken seriously many of the concerns and suggestions from both reviewers. This has led to a significantly improved manuscript, especially regarding the methods description. In my opinion, it now meets the quality standards of a scientific journal. Nevertheless, I still do not consider it interesting to a broader audience or sufficiently groundbreaking for a high impact journal. The authors have completely ignored my previous recommendations reproduced below:

"Overall, I feel that imported insights have been gained from this study, but the results and conclusions should be presented in a different way. A new manuscript should focus more on platform-related biases in data, signal standardization methods and performance measures for benchmarking. These issues should be addressed in a broader context, not limited to imputation. Less weight should be given to the challenge itself. A didactic introduction and comprehensive survey of past and present ChIP-seq, DNase-seq and ATAC-seq processing methods should be included in order to make a new paper accessible and interesting to a broader readership."

I therefore must recommend rejection of the paper in its present form.

Reviewer 2

The authors have carefully addressed almost all of my previous concerns. I only have one minor remark regarding the amended version of the manuscript.

1. In response to my previous comment 10, the authors stated that the "calligraphic R is the standard nomenclature for real-valued numbers". I consider "blackboard bold" (characters with double lines; LaTeX \mathbb{R}) the more common nomenclature for sets, including real-valued numbers. As the authors themselves use this nomenclature in the new section "2.6 New Performance Measures", I strongly recommend to use the "blackboard bold" notation/nomenclature consistently throughout the manuscript, including section 2.4.

Authors Response

Point-by-point responses to the reviewers' comments:

In response to Reviewer 1, we would reiterate that a comprehensive survey of ChIP-seq, DNase-seq, and ATAC-seq processing is outside the scope of this paper and would not enhance the findings from our challenge. It was unclear how to incorporate a survey of a variety of methods, most of which we did not use, into the narrative of us describing our challenge. However, we have included a new paragraph at the end of the discussion mentioning cross-sample normalization methods and emphasizing that normalizing these signals is a crucial part of comparative analysis.